# African Trypanosomiasis: Extracellular Vesicles Shed by *Trypanosoma brucei brucei* Manipulate Host Mononuclear Cells

**DOI:** 10.3390/biomedicines9081056

**Published:** 2021-08-20

**Authors:** Tatiana Dias-Guerreiro, Joana Palma-Marques, Patrícia Mourata-Gonçalves, Graça Alexandre-Pires, Ana Valério-Bolas, Áurea Gabriel, Telmo Nunes, Wilson Antunes, Isabel Pereira da Fonseca, Marcelo Sousa-Silva, Gabriela Santos-Gomes

**Affiliations:** 1Global Health and Tropical Medicine (GHTM), Instituto de Higiene e Medicina Tropical (IHMT), Universidade Nova de Lisboa (UNL), 1349-008 Lisboa, Portugal; tatiana.rdguerreiro17@gmail.com (T.D.-G.); joanapmarques@ihmt.unl.pt (J.P.-M.); prm.goncalves@campus.fct.unl.pt (P.M.-G.); anasbolas@gmail.com (A.V.-B.); aureamangal@gmail.com (Á.G.); mssilva.ufrn@gmail.com (M.S.-S.); 2Centro de Investigação Interdisciplinar em Sanidade Animal, Faculdade de Medicina Veterinária, Universidade de Lisboa, 1300-477 Lisboa, Portugal; gpires@fmv.ulisboa.pt (G.A.-P.); ifonseca@fmv.ulisboa.pt (I.P.d.F.); 3Microscopy Center, Faculty of Sciences, University of Lisbon, Campo Grande, 1749-016 Lisboa, Portugal; telmonunes@hotmail.com; 4Unidade Militar Laboratorial de Defesa Biológica e Química (UMLDBQ), Laboratório de Imagem Nano-Morfológica e Espectroscopia de Raios-X, 1100-471 Lisboa, Portugal; antunez.wdta@gmail.com; 5Centro de Ciências da Saúde, Departamento de Analises Clínicas e Toxicológicas, Universidade Federal do Rio Grande do Norte, Natal 59078-970, Brazil

**Keywords:** African trypanosomiasis, *Trypanosoma brucei brucei*, exosomes, macrophages, T lymphocytes, regulatory T cells

## Abstract

African trypanosomiasis or sleeping sickness is a zoonotic disease caused by *Trypanosoma brucei*, a protozoan parasite transmitted by *Glossina* spp. (tsetse fly). Parasite introduction into mammal hosts triggers a succession of events, involving both innate and adaptive immunity. Macrophages (MΦ) have a key role in innate defence since they are antigen-presenting cells and have a microbicidal function essential for trypanosome clearance. Adaptive immune defence is carried out by lymphocytes, especially by T cells that promote an integrated immune response. Like mammal cells, *T. b. brucei* parasites release extracellular vesicles (TbEVs), which carry macromolecules that can be transferred to host cells, transmitting biological information able to manipulate cell immune response. However, the exact role of TbEVs in host immune response remains poorly understood. Thus, the current study examined the effect elicited by TbEVs on MΦ and T lymphocytes. A combined approach of microscopy, nanoparticle tracking analysis, multiparametric flow cytometry, colourimetric assays and detailed statistical analyses were used to evaluate the influence of TbEVs in mouse mononuclear cells. It was shown that TbEVs can establish direct communication with cells of innate and adaptative immunity. TbEVs induce the differentiation of both M1- and M2-MΦ and elicit the expansion of MHCI^+^, MHCII^+^ and MHCI^+^MHCII^+^ MΦ subpopulations. In T lymphocytes, TbEVs drive the overexpression of cell-surface CD3 and the nuclear factor FoxP3, which lead to the differentiation of regulatory CD4^+^ and CD8^+^ T cells. Moreover, this study indicates that *T. b. brucei* and TbEVs seem to display opposite but complementary effects in the host, establishing a balance between parasite growth and controlled immune response, at least during the early phase of infection.

## 1. Introduction

African trypanosomiasis (AT), also known as sleeping sickness in humans and Nagana in cattle, is a vector-borne disease caused by an extracellular kinetoplastida parasite of the Trypanosomatidae family, genus *Trypanosoma,* and species *Trypanosoma brucei*, which is transmitted by the hematophagous dipteran *Glossina* spp. (tsetse fly). This parasitosis is considered a neglected tropical disease restricted to the intertropical region of Africa following the geographical distribution of its vector. Parasite transmission to mammals occurs by the inoculation of metacyclic trypomastigotes forms during the insect blood meal.

After being introduced into the host dermis, parasites start to replicate, giving origin to the initial lesion or the inoculation chancre [1,2]. After two or three weeks, the chancre tends to disappear, and the disease can evolve in two distinct successive phases [2]: Phase I or the hemolymphatic stage is characterized by successive waves of invasion of the blood and lymphatic system by trypanosomes, which causes intermittent fever [2,3] and phase II or meningoencephalitis stage, where the typical symptoms of sleeping sickness (dementia, cachexia, coma, and death) may become evident [2,3,4]. *Trypanosome* parasites have a dense surface coat constituted by the variant surface glycoprotein (VSG). These immunogenic coats present at the cell surface as homodimers and anchored in the membrane through glycosylphosphatidylinositol constitute the pathogen-associated molecular patterns (PAMPs) that are recognized by pattern-recognition receptors (PPRs) of innate immunity [2,4].

The innate immune response includes macrophages (MΦ), which can engulf foreign antigens through endocytosis. When soluble parasite factors and VSG interacts with MΦ, they became classically activated (M1-MΦ) and synthesize reactive oxygen intermediates (ROI), produce nitric oxide (NO), and release proinflammatory cytokines, as is the case of tumor necrosis factor (TNF)-α and interleukin (IL)-6 (type I cytokines) [2,5]. To reduce inflammation, trypanosome blood forms release some components, such as *Trypanosoma brucei*-derived kinesin-heavy chain (TbKHC-1) and adenylate cyclase (ADC), which upregulate the IL-10 production that prevents TNF−α expression [6,7], leading to the differentiation of alternative activate MΦ (M2-MΦ) and production of anti-inflammatory cytokines [5]. The inflammatory response is critical for parasite control in the early stage of infection [5] and the switch from M1-MΦ to M2-MΦ seems to occur four weeks after infection when the patient is in the late stage of disease [8].

Together with dendritic cells, MΦ are antigen-presenting cells (APC), establishing a bridge with adaptive immunity. Parasite antigens complexed with class I molecules of major histocompatibility complex (MHCI) are presented to CD8^+^ T cells and parasite antigen bound to class II molecules of major histocompatibility (MHCII) complex are recognized by CD4^+^ T cells. Both CD4^+^ and CD8^+^ T cells are crucial in the orchestration of host adaptive immune response against *T. b. brucei* parasites. However, the successive peaks of parasitemia exhibiting specific VSG induce a continuous activation of T cell clones, which leads to cell exhaustion and then to immunosuppression, and consequently impaired parasite control [9].

Like mammal cells, trypanosomes also secrete extracellular vesicles (EV), which comprise nanovesicles that differ in size and carry parasite molecular components, such as proteins, lipids, and nucleic acids [10]. The parasite *T. brucei* releases EV (TbEVs) that seems to play a role in both parasite-parasite and parasite-host interactions, influencing host immune response [11]. Moreover, TbEVs appear to incorporate different flagellar proteins, acting like virulence factors, and be highly enriched in oligomers, such as tetraspanins (a broadly expressed superfamily of transmembrane glycoproteins), known to be involved in antigen presentation, T cell signalization and activation, and in MHCI and MHCII generation. Also, it was demonstrated that upon fusion with erythrocytes, these TbEVs are responsible for the removal of red cells from the host bloodstream [12].

For parasite survival in the mammal host and success in being transmitted to the insect vector, ensuring the completion of the *T. brucei* life cycle, the infected host must develop a balanced immune response able to prevent the host killing and allow parasite replication. Despite all the studies done to understand the immune mechanisms associated with this parasitic disease, the exact role played by TbEVs in host immune response is still evasive. Thus, this work explores the effect of TbEVs in the immune activation of mouse mononuclear cells.

## 2. Materials and Methods

### 2.1. Experimental Design

To explore the effect of TbEVs on host immune response an experimental design comprising three main steps was established. The first step aims to examine the shape, size, and density of purified TbEVs using electron microscopy and nanoparticle tracking analysis. In the second step, the aim was to investigate the innate activity of MΦ when stimulated by TbEVs, including their microbicide and antigenic presentation potential, and the last goal step aims to assess the differentiation of T cell subsets related to effector and regulatory adaptive immune response.

Mouse MΦ-like cells were exposed to *T. b. brucei* trypomastigotes and stimulated by TbEVS and BALB/c mice peripheral blood mononuclear cells (PBMC) were exposed to *T. b. brucei* trypomastigotes and stimulated by TbEVS and parasite antigen (Ag). Type I and type II MΦ activity was analyzed by NO and urea production using colourimetric assays. The potential of MΦ to present parasite antigens to lymphocytes was indirectly evaluated by the expression and density of surface molecules MHCI and MHCII by flow cytometry. T lymphocyte subsets were immunophenotyped by flow cytometry through surface expression and density of CD3 and CD25 molecules and intracellular expression of FoxP3. In parallel, resting cells, MΦ stimulate by phorbol myristate acetate (PMA) and PBMC stimulated by concanavalin A (ConA) were also evaluated. Furthermore, obtained data were subject to statistical procedures.

### 2.2. BALB/c Mice

Six to eight-week-old male BALB/c *Mus musculus* mice were purchased from the Instituto Gulbenkian de Ciências (IGC, Lisbon, Portugal) and maintained in the IHMT animal facility, in sterile cabinets with sterile food and water ad libitum. Mice were used to recover *Trypanosome* blood forms and to isolate PBMC. The animals were handled according to the Portuguese National Authority for Animal Health (Ref. 0421/000/000/2020, 23 September 2020, DGAV—Direção Geral de Alimentação e Veterinária), in conformity with the institutional guidelines and the experiments performed in compliance with Portuguese law (Decree-Law 113/2013), EU requirements (2010/63/EU), and following the recommendations of the Federation of European Laboratory Animal Science Associations (FELASA).

### 2.3. Trypanosoma brucei brucei Parasites

Peripheral blood of BALB/c mice infected with *T. brucei brucei* strain G.V.R. 35 was collected (Appendix A) and treated with ammonium-chloride-potassium lysis buffer. *Trypanosoma* blood forms were then purified using diethyl aminoethyl (DEAE)-cellulose columns [13] and maintained in Schneider Drosophila medium (Sigma-Aldrich, Hamburg, Germany) supplemented with 10% (*v*/*v*) of heat-inactivated fetal bovine serum (hiFBS) free of extracellular vesicles (FBS-exofree, Thermo Fisher Scientific, Waltham, MA, USA) at 24 °C. Parasite morphology and motility were checked every day by direct microscopy and, parasite topography was evaluated by scanning electron microscopy (SEM, SEM-UR- LBDB Hitachi SU8010 High-Technologies Corporation, Ibaraki, Japan). TbEVs were purified from the supernatant of cultures exhibiting motile trypomastigotes.

### 2.4. T. b. brucei Extracellular Nanovesicles and Parasite Crude Antigen

Trypomastigotes harvested from cultures by centrifugation at 2000× *g* for 10 min at 4 °C were used to produce parasite antigen (Ag), and TbEVs were obtained from culture supernatants. Parasites were washed twice in phosphate-buffered saline (PBS) 2 mM ethylenediaminetetraacetic acid (EDTA), resuspended in PBS, and then disrupted by eight freeze-thawing cycles ranging from −20 °C to room temperature. After centrifugation, protein content (mg·mL^−1^) was determined using a Nanodrop 1000 spectrophotometer (Thermo Scientific, Waltham, MA, USA), and Ag was preserved at −20 °C. Supernatants of trypomastigote culture incubated for 48 h were filtered through 0.2 mm syringe filters (VWR International, Radnor, PE, USA), and TbEVs were purified using Exosome Spin Columns (Invitrogen, Waltham, MA, USA) according to the manufacturer’s instructions. Isolated TbEVs were examined by SEM, single particle size and concentration analyzed by Nanoparticle Tracking Analysis (NanoSight NTA 3.2), and protein content was estimated by spectrophotometry.

### 2.5. Morphology of T. b. brucei and Parasite Extracellular Vesicles

Cultured *T. b. brucei* parasites deposited on glass slides were observed under an optical microscope to evaluate parasite motility and morphology. The topography of TbEVs and morphology of cultured-derived parasites were examined by SEM.

Round glass coverslips were immerged into poly-D-lysine (Sigma-Aldrich, Burlington, MA, USA) overnight to increase adherence and later placed in a 24-well plate. Then parasites were allowed to adhere to the coverslips and were fixed with PBS 4% paraformaldehyde (Merck, Rahway, NJ, USA) for 30 min at 4 °C. Coverslips were rinsed three times with distilled water, treated with 0.5% osmium tetroxide (Sigma-Aldrich), and washed again. Then, plates were incubated with a fixative solution of 1% tannic acid (Sigma-Aldrich) for 30 min. TbEVs were fixed to coverslips with 2.5% glutaraldehyde, 0.1 M sodium cacodylate buffer, pH 7.4 for 2 h at 4 °C. Afterwards, both parasites- and TbEV-coverslips were washed and then dehydrated by sequential addition of 30%, 50%, 70%, 80%, and 90% ethanol for 5 min each. Coverslips were immersed in 100% ethanol and then treated with hexamethyldisilazane solvent (Sigma-Aldrich), coated with gold-palladium, mounted on stubs to be observed under an ultra-high resolution scanning electron microscope. The surface area and perimeter of TbEVs were quantified from acquired images using Image J software.

### 2.6. Macrophage Cell Line and Primary Mononuclear Cells

PMBC were isolated from healthy (non-infected) BALB/c mice by density gradient [14]. Briefly, cells at Hystopaque-1077 (Sigma-Aldrich) and plasma interface were removed and washed three times in PBS by centrifugation at 370× *g* for 10 min at 4 °C. Then, the supernatant was discarded and the PMBC-containing pellet resuspended in PBS. Cell viability was evaluated using the trypan blue staining method and cell concentration estimated in a Neubauer-counting chamber by direct microscopy.

A macrophage (MΦ)-like cell line (P388D1, ATCC, Bird Park, Hawaii, USA) previously isolated from a mouse lymphoma was expanded in RPMI-1640 (Lonza, Basel, Switzerland) supplemented with 10% (*v*/*v*) of hiFBS, 2 mM L-glutamine (Merck) (complete RPMI medium) at 37 °C in a humidified atmosphere with 5% CO_2_.

### 2.7. Stimulation of Macrophages and PBMC by TbEVs

MΦ (2 × 10^6^ cells per well) and PBMC (1 × 10^5^ cells per well) were plated in a sterile 96-well plate with complete RPMI medium and separately exposed to viable parasites (3 parasites per cell) and stimulated with TbEVs (10 μg·mL^−1^). PBMC also was stimulated by *T. b. brucei* soluble antigen (10 μg·mL^−1^). MΦ plates were incubated for 24 h and PBMC plates for 72 h at 37 °C in a humidified atmosphere with 5% CO_2_. In parallel, unstimulated cells, used as the negative control, and PMA (Promega, Madison, WI, USA) stimulated MΦ and ConA (Sigma-Aldrich)-stimulated lymphocytes, used as the positive controls, were also incubated.

### 2.8. Nitric Oxide and Urea Production by Macrophages

Supernatants of MΦ exposed to viable parasites, stimulated by TbEVs and PMA, and non-stimulate (resting MΦ) were collected and used for quantification of urea by the commercial kit QuantiChrom™ Urea Assay Kit-DIUR-100 (BioAssay System, Hayward, CA, USA) and the indirect measurement of NO levels through the detection of NO_2_^−^ and NO_3_^−^ using the Nitrate/Nitrite Colorimetric Assay (Abnova, Walnut, CA, USA). Both assays were performed according to the manufacturer’s instructions.

### 2.9. Macrophage and Lymphocyte Immunophenotyping

Cells (MΦ and PBMC) exposed to motile parasites and stimulated were harvested from plates and washed with PBS. MΦ were labelled with mouse anti-MHCI (H-2Kb) monoclonal antibody FITC directly conjugated and mouse anti-MHCII (I-A/I-E) monoclonal antibody PE directly conjugated (BioLegend, San Diego, CA, USA).

PBMC were resuspended in PBS 0.5% hiFBS 2 mM EDTA and were added magnetic microbeads coated with mouse anti-CD8a (Ly-2) monoclonal antibody (Miltenyi Biotec, Bergisch Gladbach, Germany). Cells were incubated for 15 min at 4 °C protected from light and then washed. CD8^+^ cells were sorted by positive selection, using a MACS^®^ system (Miltenyi Biotec) while CD8^−^ cells were eluted.

Sorted CD8^+^ and CD8^−^ cell fractions were washed with PBS 2% hiFBS 0.01% NaN_3_ and incubated for 30 min with mouse anti-CD3 monoclonal antibody FITC-directly conjugated and mouse anti-CD25 monoclonal antibody PerCP-cy5.5 directly conjugated (BioLegend). After two washes in PBS, cell suspensions were fixed with PBS 2% formaldehyde. Cells were then resuspended in permeabilization buffer (PBS 1% FBS, 0.1% NaN3, 0.5% Triton-X, pH 7.4–7.6) and incubated for 20 min, washed and labelled with mouse anti-FoxP3 monoclonal antibody PE directly conjugated (BioLegend).

Cell acquisition was performed in a 4-colour flow cytometer (BD FACSCalibur, BD Biosciences, USA) and data were analyzed using Flowjo V10 (Tree Star Inc., Ashland, OR, USA). FSC-H vs. SSC-H gate was used to remove debris and pyknotic cells in the lower left quadrant of the plot as well as the large (off-scale) debris found in the upper right quadrant. Singlet gate was used to define the non-clumping cells based on pulse geometry FSC-H vs. FSC-A, eliminating the doublets. CD3^+^, CD25^+^, and FoxP3^+^ cell subsets were defined using fluorescence minus one control (FMOs).

To validate the composition of magnetically separated cell fractions, samples of unprimed CD8*^+^* and CD8^−^ cell fractions were stained with the anti-CD3 monoclonal antibody and anti-CD4 monoclonal antibody PerCp directly conjugated (BioLegend) and evaluated by flow cytometry. In CD8^+^ cell fraction, less than 20% of cells presented a CD3^+^CD8^−^ phenotype, indicating that this cell fraction mainly was constituted by CD8^+^ T cells and in the CD8^−^ cell fraction ≈85% of the cells evidenced a CD3^+^CD4^+^ phenotype, which is consistent with the predominance of CD4^+^ T cells.

### 2.10. Statistical Analysis

Data analysis was performed using GraphPad Prism version 8 (GraphPad Software, San Diego, CA, USA). After verification by a Kolmogorov–Smirnov test that the data of the current study do not evidence a normal distribution, significant differences were determined using the non-parametric Wilcoxon matched-pair signed-rank test. A 5% (*p* < 0.05) significance level was used to evaluate statistical significance. The surface and perimeter of TbEVs are represented by violin plots (median, interquartile ranges and distribution). Cell results of at least three independent experiments evaluated in triplicate are expressed by whiskered box-plots, indicating the median, maximum and minimum values or by graph bars (mean and standard error). The relative importance of TbEVs in cell activity was assessed by the principal component analysis (PCA) of exploratory multivariate statistical analysis, using Past4.03 (Natural History Museum, University of Oslo, Oslo, Norway). PCA analysis organizes data in principal components, which can be visualized graphically with a minimal loss of information, making visible the differences between the activation of cells exposed to *T. b. brucei* parasites, TbEVs and Ag stimulated cells. K-means cluster analysis also was used for cluster validation.

## 3. Results

### 3.1. T. b. brucei Trypomastigote Forms Release Extracellular Vesicles

Cultured-derived *T. b. brucei* parasites observed by direct microscopy and by scanning electronic microscopy showed an elongated body, a flagellum (Figure 1A), and an undulating membrane (Figure 1B), which are recognised as the morphological characteristic of the trypomastigote form. Besides, cultured-derived parasite exhibited nanovesicles that seems to bud from the cell surface and flagellum (Figure 2A,B).

Suspension of purified nanovesicles derived from trypomastigote forms of *T. b. brucei* evidenced a mean protein concentration of 5.107 μg·mL^−1^ (≈35.28) and NTA analysis revealed that the size of TbEVs was within a range of 50 nm and 350 nm.

Two peaks of higher concentration of TbEVs with sizes around 100 and 170 nm were detected, while TbEVs bigger than 200 nm seem to be rare (Figure 2C). The topographic analysis place in evidence vesicles with a spherical shape and a smooth surface (Figure 3A), with perimeter ranging between 335 and 3311 nm (mean 1634 nm ± 94.07) (Figure 3B) and surface area between 9 and 872 nm (mean 255.2 nm ± 25.40) (Figure 3C). These data also indicate that cultured-derived *T. b. brucei* parasites release two main classes of TbEVs: small TbEVs (with perimeter and surface area around 65 nm and 905 nm, respectively) and large TbEVs (with perimeter and surface area around 346 nm and 2088 nm, respectively).

### 3.2. T. b. brucei and TbEVs Induced Mouse MΦ to Produce NO and Urea

MΦ activity after stimulation with TbEVs was examined by the ability of cells to metabolize arginine (Figure 4).

Resting-MΦ incubated for 24 h (negative control) exhibited a residual NO synthesis (Figure 4B). However, after PMA-stimulation (positive control), cells revealed a significant NO increase (*p* = 0.0078), thus confirming the viability and functionality of these cells that could transform arginase into NO through the enzymatic activity of NOS2. MΦ stimulated by TbEVs (*p* = 0.0313) or *T. b. brucei* parasites (*p* = 0.0078) (Figure 4A) showed significant increases in NO levels when compared with resting MΦ. MΦ exposed to parasites exhibited the higher NO production that was significantly different from TbEVs exposed-MΦ (*p* = 0.0313).

In resting-MΦ were detected low levels of urea (Figure 4C). However, PMA stimulated cells showed a significant increase in urea production (*p* = 0.0156), indicating that these cells can convert arginine into urea through arginase enzymatic activity. MΦ stimulated by TbEVs or exposed to parasites also exhibited significant high levels of urea when compared with resting-MΦ (*p* = 0.0313).

Although both *T. b. brucei* parasites and TbEVs promote mouse MΦ to produce NO and release urea, parasites elicit the highest secretion.

### 3.3. TbEVs Direct the Differentiation of MHCI^+^, MHCII^+^ and MHCI^+^MHCII^+^ Macrophage Subsets

To indirectly evaluate the possible presentation of parasite antigens by mouse MΦ, the expression of MHC molecules by MΦ exposed to TbEVs for 24 h and the frequency of MHCI^+^ and MHCII^+^ MΦ subsets were analyzed by flow cytometry (Supplementary Appendix A and Figure 5).

A significant high frequency of MHCI^+^ (Figure 5A), MHCII^+^ (*p* = 0.0001) (Figure 5B), and MHCI^+^MHCII^+^ MΦ (*p* = 0.0313) was observed in cells stimulated by PMA (Figure 5C) when compared with resting cells. MHCI^+^ MΦ (*p* = 0.01) and MHCII^+^ MΦ (*p* = 0.0039) subsets were inhibited after *T. b. brucei* exposure. On the other hand, TbEVs elicited a significantly high expression of MHCI^+^ (*p* = 0.0003) and MHCII^+^ MΦ (*p* = 0.0004) in comparison to resting cells and cells exposed to parasites (*p* _MHCI_^+^ _MΦ_ <0.0001, *p* _MHCII_^+^
_MΦ_ = 0.0039).

TbEVs seemed to be responsible for a significative expansion of MHCI^+^MHCII^+^ MΦ subset (*p* = 0.0313) (Figure 5C). Also, in this case, the exposure to parasites caused a significant reduction of this cell subpopulation when comparing with resting-MΦ and TbEVs stimulated MΦ (*p* = 0.0313).

When analyzing the levels of fluorescence intensity, no differences were observed between resting-MΦ, MΦ exposed to parasites and TbEVS stimulated MΦ. Only PMA-stimulated MΦ evidence a considerable augment of MHCI (*p* = 0.0313).

Altogether the results indicated that TbEVs enhance the expression of MHCII class I and class II in MΦ, favouring antigen presentation, but did not increase the surface density of these molecules on the cell surface. On the other hand, *T. b. brucei* parasites promote a reduction of MHCI^+^, MHCII^+^ and MHCI^+^MHCII^+^ MΦ subsets, possible avoiding the antigen presentation and activation of cytotoxic and helper T cells.

### 3.4. TbEVs Promoted Specific Mouse Macrophage Activation

To identify correlations between the influence of TbEVs and *T. b. brucei* parasites on mouse MΦ, data were analysed by PCA. This statistical analysis indicated that the overall influence of TbEVs and PMA on MΦ activity is correlated (Figure 6A). On contrary, *T. b. brucei* effects on mouse MΦ were distinct from TbEVs and resting-MΦ. These results were confirmed by cluster analysis, which aggregates in the same cluster (cluster 3) the effects of TbEVs and PMA on MΦ. MΦ activity caused by parasite exposure was found through clusters 2 and 3, with most of the effects in cluster 2 whereas resting-MΦ is mainly localized in cluster 1 (Figure 6B).

Therefore, this analysis highlights that TbEVs activation of rodent MΦ can be different from cell activation induced by *T. b. brucei*.

### 3.5. TbEVs Favored the Differentiation of CD4^+^ T Cells and Increment the Surface Expression of CD3 Molecules

The frequency of CD3^+^ cells (Figure 7) and expression level of CD3 molecules were examined in mononuclear blood cells exposed to parasites or stimulated by TbEVs or parasite Ag (Appendix A).

In both CD4^+^ (Figure 7A) and CD8^+^ (Figure 7B) cell fractions, the frequency of CD3^+^ cells were significantly lower in cells exposed to *T. b. brucei* in comparison to unprimed cells (*p* = 0.0078). However, ConA stimulation caused a significant increase in CD3^+^ cells in both cell fractions (*p* = 0.0078). In both cell fractions, cells exposed to parasites were also statistically different from TbEVs and Ag stimulated cells (*p* = 0.0313). TbEVs were responsible for a significant increase of CD3^+^ cell subset in the CD4^+^ cell fraction (*p* = 0.0078) in comparison with unprimed and parasite exposed cells.

The fluorescence intensity of CD3-labelled cells in both cell fractions significantly increased (*p* = 0.0313) in Ag and TbEVs stimulated PBMC when compared to unprimed cells. Stimulation by ConA also induced a significant increase (*p* = 0.0313) in CD3-fluorescence intensity of CD4^+^ cell fraction. On the other hand, when compared to unprimed cells and TbEVs and Ag-stimulated cells, exposure to *T. b. brucei* cultured-derived parasites weakened CD3 fluorescence in CD4^+^ and CD8^+^ cell fractions (*p* = 0.0313).

Altogether, these results indicated that TbEVs stimulation seemed to trigger the expansion of the CD4^+^ (CD3hi) T cell subset and induce the expression of CD3 molecules on CD8^+^ T cells. In contrast, *T. b. brucei* parasites impaired the differentiation of CD8^+^ and CD4^+^ T cells and promoted the down expression of CD3 molecules at the cell surface.

### 3.6. TbEVs Led the Expansion of Regulatory CD4^+^ T Cells and FoxP3^+^CD4^+^ T Cell Subset and Enhanced Surface CD25^+^ and Intracellular FoxP3^+^ Molecules

To assess the effect of TbEVs on CD4^+^ and CD8^+^ T cell subsets, mononuclear blood cells were exposed to cultured parasites and stimulated by TbEVs or parasite Ag. The frequency of CD4^+^ T cells expressing CD25 (Figure 8) molecules was evaluated as well as the density of CD25 molecules on the cell surface and intracellular FoxP3 molecules on stimulated cells. Since parasite-exposed cells evidence a low frequency of CD3^+^ cells only expression and density CD25 were examined.

*T. b. brucei* parasites induced a significant expansion of the CD4^+^ T cell subset expressing CD25 comparing to unprimed cells, TbEVs, and Ag stimulated cells (*p* = 0.0313) (Figure 8C). Parasite Ag and TbEVs caused a high expansion of FoxP3^+^ CD4^+^ (Figure 8D) and FoxP3^+^CD25^+^ CD4^+^ T cells (*p* = 0.0313) (Figure 8A) in comparison to unprimed cells.

Stimulation of PBMC by ConA led to a significant expansion (*p* = 0.0313) of CD4^+^ T cell subset (Figure 8B). On the other hand, parasites, TbEVs or Ag did not seem to affect the CD4^+^ (FoxP3^−^ CD25^−^) T cell subset.

TbEVs and parasites promoted a higher fluorescence intensity of CD25-labeled cells when compared with unprimed lymphocytes (*p* = 0.0313). Furthermore, *T. b. brucei* parasites induced a higher expression of CD25 molecules in comparison to TbEVs and Ag stimulated cells (*p* = 0.0313). Significant overexpression of intracellular FoxP3 molecules was also found in CD4^+^ T cells after stimulation with TbEVs or Ag (*p* = 0.0331).

These results indicated that TbEVs and parasite Ag favoured expansion of regulatory CD4^+^ T cell subset (FoxP3^+^CD25^+^CD3^+^ CD4^+^, T reg cells) and FoxP3^+^CD4^+^ T cells. Moreover, *T. b. brucei* parasites enhanced the frequency of CD25^+^CD4^+^ T cells. Among CD4^+^ T cells, the overexpression of α-chain of IL2-receptor (CD25) was induced by parasites and TbEVs, and the increased expression of FoxP3 was elicited by TbEVs and Ag.

### 3.7. TbEVs Direct the Expansion of Effector CD8^+^ T Cells and CD8^+^ T Cells Expressing FoxP3^+^ Phenotype and Increase Surface CD25 and Intracellular FoxP3 Molecules

CD8^+^ T cell subsets were examined by estimating the frequency of CD8^+^ T cells expressing CD25 in mononuclear blood cells exposed to parasites and stimulated by TbEVs or parasite Ag (Figure 9). The expression of FoxP3 was evaluated in stimulated cells. Furthermore, the potential of these cells to recognize IL-2 and regulate the cell immune activity was indirectly assessed by the density of surface CD25 and intracellular FoxP3 molecules.

When compared to unprimed CD3^+^CD8^+^ cells, it was observed that ConA stimulation caused a significant high frequency of CD8^+^ T cell subsets expressing CD25^+^ and FoxP3^+^ (*p* _CD25_^+^
_FoxP3_^+^ = 0.0313, Figure 9A; *p* _CD25_^+^ = 0.0078, Figure 9C) as well as CD8^+^ (CD25^−^FoxP3^−^) T cells (*p* = 0.0516, Figure 9B). Cells stimulated by TbEVs (*p* = 0.0156) showed significant high frequencies of CD25^+^FoxP3^+^CD8^+^ T cells (Figure 9A).

TbEVs and Ag induced a significant expansion of CD8^+^ (CD25^−^FoxP3^−^) T cell subset (*p* = 0.0156) and promoted the retraction of CD25^−^FoxP3^+^ CD8^+^ T cell subset (*p* = 0.0313) (Figure 9B). However, when compared to unprimed cells, *T. b. brucei* parasites caused a significant restrain of CD8^+^T cells.

CD8^+^ T cells exposed to *T. b. brucei* parasites or stimulated by ConA, TbEVs, and parasite Ag showed a significant increase of CD25 molecules (*p* = 0.0313) when comparing with unprimed cells. Furthermore, FoxP3 fluorescent intensity also increased in CD8^+^T cells stimulated by TbEVs (*p* = 0.0313).

Altogether the results indicated that TbEVs mainly triggered the expansion of effector CD8^+^T cells and CD8^+^ T cell subsets expressing FoxP3 associated with a higher density of FoxP3 molecules. The overexpression of α-chain of IL2-receptor (CD25) induced by parasites can be mainly associated with the expansion of CD25^+^ CD8^+^ T cell subset.

### 3.8. TbEVs and Parasite Ag Triggered Related Influence on the Phenotype of BALB/c Mice T Cells

PCA analysis of CD4^+^ and CD8^+^ T cells indicated a positive correlation between the effect of TbEVs and parasite Ag on T cells (Figure 10A), which contrasted with *T. b. brucei* effects. These results were confirmed by cluster analysis, which showed the effects of Ag and TbEVs grouped in cluster 3. On the other hand, the influence of parasites in T cells was grouped at cluster 1. Furthermore, ConA stimulation outcomes were localized in cluster 2 and the intrinsic activity of unprimed T cells were mainly within cluster 3 (Figure 10B). Overall, the effect of TbEVs on BALB/c mice T cells seemed to be similar to cell stimulation caused by parasite Ag and slightly different of unprimed cells, but completely different from *T. b. brucei* parasites.

## 4. Discussion

African trypanosomes have been extensively studied since they are responsible for severe diseases in both medical and veterinary contexts. To survive, this extracellular protozoan affords several mechanisms that use to evade the mammals’ immune system. Also, to ensure the completion of its life cycle, the parasite needs to avoid host mortality during the hemolymphatic phase. Therefore, a balance between infection level (parasitemia) and the intensity of the inflammatory immune response must be achieved in the host. Since safe and efficient anti-trypanosomal drugs and vaccines are lacking, many are the studies performed to understand the mechanisms associated with the host immune response direct against *T. brucei* infection. Although different approaches were applied, and numerous findings reported, some questions remain unanswered, and some mechanisms are not entirely understood or still controversial.

Since EVs released by eucaryotic cells seem to play a role in intercellular communication, interfering with several cellular processes, such as the activation of microbicide processes and generation of immune mediators by changing gene expression and affecting signalling pathways the potential of trypomastigotes derived EVs in influencing the immune activity of innate and adaptive immune cells was explored. Findings of the current study indicate that TbEVs shed by trypomastigotes is enriched in proteins and are a heterogeneous population of spherical vesicles mainly constituted by smaller (<0.17 nm) and biggest (>0.17 nm) EVs, which is in line with previous findings [12].

Since the expansion in different organs and tissues of the MΦ population, as is the case of the liver, spleen, and the bone marrow was described in *T. brucei* infected mice, the intercommunication of TbEVs with innate immune cells were examined. It is reported that inducible nitric oxide synthase (iNOS) peaked six days post-infection and that oxidized l-arginine generates l-citrulline and NO [15]. In the current study, TbEVs trigger mouse MΦ to produce NO as well as *T. brucei* cultured parasites, suggesting a role for TbEVs at the early stage of sleeping sickness. NO is a highly regulated effector molecule, which participates in several physiological and immune processes and can inactivate pathogens including *T. brucei* parasites [15]. However, high NO levels can be a disadvantage to the host, given the large spectra of cell disorders that are associated with NO activity. In sleeping sickness, the persistence of M1-MΦ is related to anaemia and systemic immune response [16,17,18,19].

To counteract the possible harmful effect of NO in the host and perpetuate parasite replication, African trypanosomes induce the differentiation of M2-MΦ. L-Arginine can be hydrolyzed by arginase, generating ornithine, urea, proline and polyamines [20]. A previous study performed in *T. b. brucei*-infected mice indicates that M1-MΦ predominate in the early stage of infection while M2-MΦ prevail at a later infection stage (1–4 weeks) [18]. In the current study was found that TbEVs and cultured parasites induced urea and NO production by mouse MΦ. These findings reinforce the role of TbEVs in the early stage of infection as a MΦ modulator. By inducing the differentiation of mouse MΦ expressing M1 and M2 phenotypes, TbEVs seem to strengthen parasite activity in sustaining a balance between pro-inflammatory and anti-inflammatory responses. Moreover, by simultaneously ensuring the supported production of polyamines, which are essential nutrients for parasite survival and replication [7,21], TbEVs can facilitate the establishment of a chronic infection. The kinesin-1 heavy chain (TbKHC-1) released by African trypanosomes has been described as an activator of the arginine pathway, leading to the production of polyamines [7]. Taking into account the above considerations, TbKHC-1 may form part of TbEVs’ cargo.

Contrary to *T. b. brucei* parasites that cause a marked reduction in the frequency of MHCI^+^ and MHCII^+^ MΦ, affecting the recognition of parasite antigens by T lymphocytes and compromising the adaptive immune response, TbEVs induce differentiation of MHCI^+^ and MHCII^+^ MΦ and double-positive MΦ (MHCI^+^ MHCII^+^), indicating that these cells can establish a link with acquired immunity by presenting parasite antigens to helper and cytotoxic T cells. Even so, the density of MHCI and MHCII molecules on the membrane surface of MΦ remained similar to resting MΦ. Though not fully understood similar inhibition of MHC expression avoiding antigen presentation was mentioned in a study carried out on *T. cruzi*, which is an intracellular parasite responsible for Chagas’ disease or American trypanosomiasis [22]. Therefore, TbEVs that appear to exert an action contrary to the parasite may cause the activation of adaptive immunity at least to some point, activating both CD4^+^ and CD8^+^ T cells.

During the course of infection, *T. brucei* releases the stumpy induction factor (TSIF) that has been associated with the impairment of T cell proliferation [23]. A previous study performed in patients infected with *T. b. gambiense* has shown that T cells were considerably lower when compared with controls [24]. Thus, considering the high importance of cellular activation in orchestrating an integrated host immune response against sleeping sickness, the communication established by TbEVs on T cell expansion was also examined. In agreement with previous findings, a marked reduction of both CD3^+^CD4^+^ and CD3^+^CD8^+^ T cells caused by *T. b. brucei* parasites was found in the current study. However, TbEVs favour the expansion of CD4^+^ T cell subpopulation and induce the increase of CD3 expression in both CD4^+^ and CD8^+^ subsets (CD3^hi^ CD4^+^ and CD3^hi^ CD8^+^ T cells). CD3 complex is a T cell co-receptor responsible for intracellular signalling. Altogether, these findings indicate that, contrary to *T. b. brucei*, TbEVs may have the necessary conditions to stimulate T cells. However, in a previous study, Boda and colleagues [25] reported that effector CD8^+^ T cells were significantly lower in patients with African trypanosomiasis which corroborates our findings.

In the current study, *T. b. brucei* parasites induce the expansion of CD4^+^ and CD8^+^ T cell subpopulations expressing a CD25 phenotype and favour the expression of CD25 molecules (CD25^hi^CD4^+^ and CD25^hi^CD8^+^ T cells). CD25 is the α chain of the IL-2 receptor, which is expressed by regulatory T (Treg) cells. CD25 is required for interleukin (IL)-2 signalling that mediates T cell activation and proliferation. To become fully activated, CD8^+^ T cells require the help of CD4^+^ T cells mediated by IL-2. However, Kalia et al. [26] have found that CD25^hi^CD8^+^ T cells can perceive proliferation signals (IL-2) and differentiate into short-lived effector cells and Olson et al. [27] reported that lymphocyte triggering factor (TLTF) released by *T. b. brucei* specifically elicit CD8^+^ T cells to generate IFN-γ. Moreover, it was reported that activated CD4^+^ T cells secrete IFN-γ and is recognised that this cytokine functions as a *T. b. brucei* growth factor [28]. Thus, *T. b. brucei* parasites may favour the expansion of CD25^+^ T cells to ensure their survival.

FoxP3 nuclear factor is a regulator of gene expression that suppresses the function of NFAT and NF-κB nuclear factors, avoiding the expression of pro-inflammatory cytokine genes, including IL-2. CD4^+^ Treg cells which constitutively express FoxP3 can suppress leukocyte effector activity, contributing to the maintenance of immune homeostasis. These regulatory cells also exhibit a high expression of CD25 molecules. In African trypanosomiasis, the expansion of Treg cells seems to occur from the parasite establishment to the chronic stage, being associated with parasite tolerance (reviewed in [29]) and in trypanotolerant C57BL/6 mice, the expansion of CD25^+^ FoxP3^+^ CD4^+^ T cell subset was demonstrated after the first peak of parasitemia [30]. On the other hand, the lack of expansion of Treg cells is associated with tissue damage and impaired survival of infected mice [31]. In the current study, TbEVs promote the expansion of CD4^+^ and CD8^+^ Treg cells and FoxP3^+^ CD4^+^ T cell subset. According to Zelenay et al. [32], CD4^+^ T cells expressing FoxP3^+^ phenotype are committed Treg cells able to regain CD25 expression. Thus, TbEVs seem to induce the differentiation of CD4^+^ and CD8^+^ Treg cells and unconventional CD4^+^ Treg cells. In previous studies, it was reported that unconventional-Treg cells also can exert regulatory functions [33]. During chronic parasite infection, uncontrolled inflammation is one of the most clinical features noticed that can become lethal if not controlled by Treg cells [34,35]. Therefore, during infection TbEVs can lead to the establishment of a pool of T cells with a regulatory phenotype able to balance excessive inflammation.

Altogether, these findings indicate that *T. b. brucei* parasites can activate the metabolization of arginine by MΦ and [36] modulate T cells, ensuring the production of polyamines and IFN-γ, both critical for parasite survival. At the same time, this parasite avoids antigen presentation by MΦ and T cells activation.

However, the findings of the present study place in evidence that TbEVs can establish direct communication with cells of innate and adaptive immunity, and the effect of TbEVs on these cells is different from the parasite itself. TbEVs can elicit parasite antigenic presentation to CD4^+^ and CD8^+^ T cells possible leading to the activation of the cellular immune response in addition to the bidirectional activation of mouse MΦ, contributing to the release of factors essential to parasite growth and, also, to parasite destruction. Moreover, TbEVs also seems to have a direct effect on CD4^+^ T lymphocytes, triggering the expression of T cell co-receptors, which are key players in intracellular signalling. The expression of the nuclear factor FoxP3 was also stimulated by TbEVs, guiding the differentiation of CD4^+^ and CD8^+^ Treg cells. imposing regulation on host inflammatory immune response which is a hallmark of sleeping sickness. Since an unbalanced inflammatory response against the parasite can become lethal to the host, the role of Treg cells in protecting from collateral tissue damage is crucial. On the other hand, TbEVs induced the differentiation of effector CD8^+^ T cells and drive the overexpression of FoxP3 molecules in CD4^+^ T cells which are involved in maintaining immune homeostasis. Therefore, parasites and TbEVs seem to display complementary effects in the host immune response that can ensure parasite survival, delaying disease severity and the eventual death of the host. Furthermore, TbEVs represent a source of biomarkers that can open new avenues for a better diagnosis of sleeping sickness and the development of prophylactic measures to control the disease caused by *T. brucei* in Subsaharan Africa.

## Figures and Tables

**Figure 1 biomedicines-09-01056-f001:**
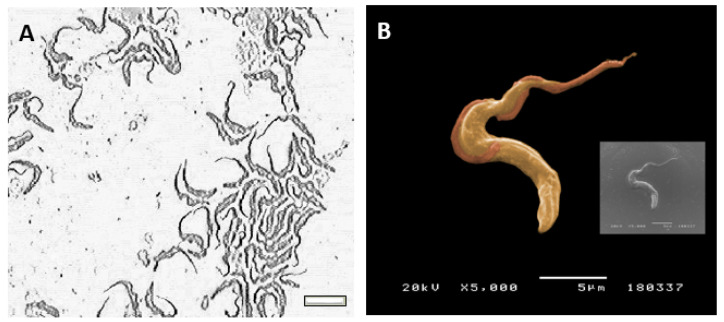
Trypomastigote forms of *Trypanosoma brucei brucei*. Culture-derived parasites were morphologically evaluated under an inverted light microscope (**A**, size bar—10 µm, ×400 magnification) and after metalization topographically analyzed by SEM (**B**). Image B was artificially coloured using the GIMP2.10.0 software.

**Figure 2 biomedicines-09-01056-f002:**
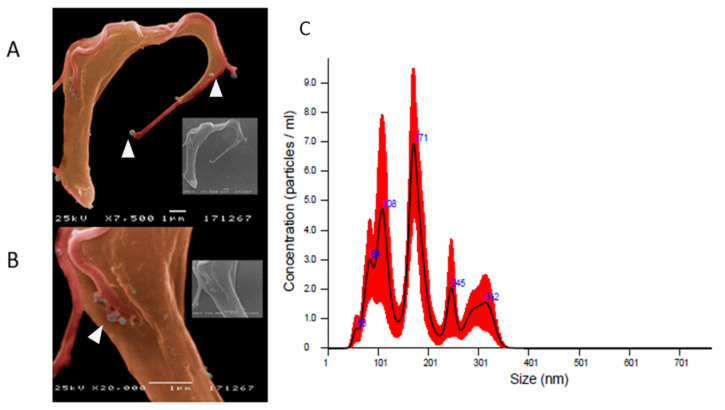
Release of extracellular nanovesicles by *T. b. brucei* and size and concentration of purified TbEVs. Representative images (**A**,**B**) of culture-derived parasites shedding nanovesicles (arrowheads) were acquired by SEM. TbEVs harvested from supernatants of *T. b. brucei* cultures were analyzed by NTA (**C**). Mean values (blue) + SD (red) are represented by average finite track length adjustment (FTLA) concentration/size graph.

**Figure 3 biomedicines-09-01056-f003:**
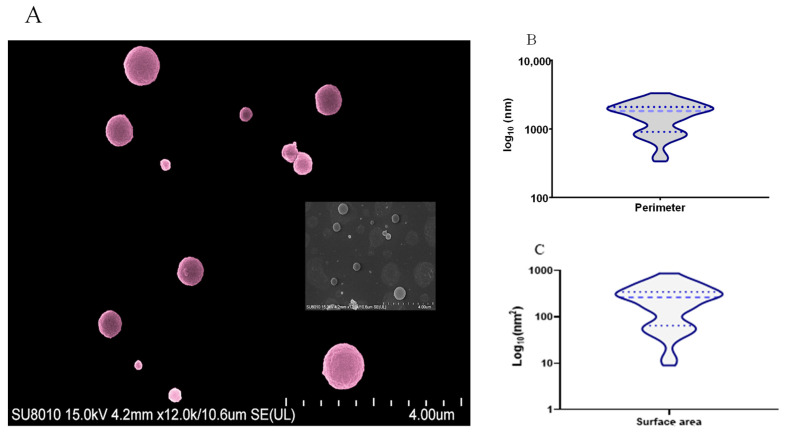
Topography and dimensions of *T. b. brucei* extracellular nanovesicles. TbEVs purified from culture-derived parasites were analyzed by SEM and images were acquired. A representative image of TbEVs (**A**) artificially coloured, using the GIMP2.10.0 software is shown. Perimeter (**B**) and surface area (**C**) of TbEVs estimated by Image J are represented by violin plots, indicating median (thick blue bars), interquartile range (blue dots) and distribution.

**Figure 4 biomedicines-09-01056-f004:**
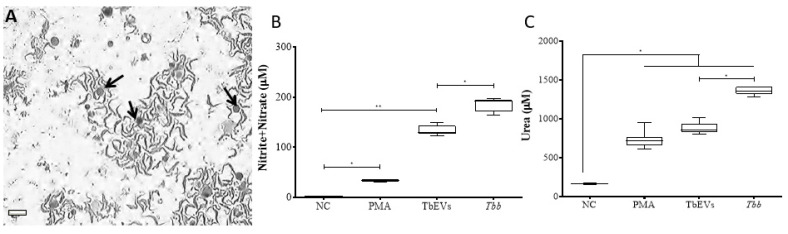
Production of nitric oxide and urea by macrophages exposed to TbEVs. MΦ (arrows) exposed to trypomastigote forms were observed under an inverted light microscope ((**A**), size bar—20 µm, ×100 magnification). NO (**B**) and urea (**C**) production were evaluated in MΦ exposed to *T. b. brucei* (*Tbb*) parasites, in MΦ stimulated by TbEVs and PMA, and in resting-MΦ (negative control, NC). Results of three independent experiments (*n* = 12) and two replicates per sample are represented by whiskered box plots, including median, minimum, and maximum values. The non-parametric Wilcoxon matched-pairs signed-rank test was used for statistical comparisons (*p* < 0.05). * (*p* < 0.05) and ** (*p* < 0.01) indicate statistical significance.

**Figure 5 biomedicines-09-01056-f005:**
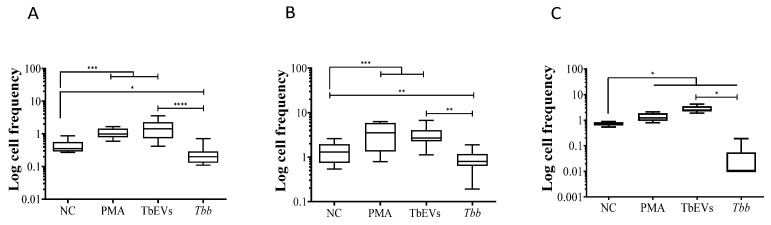
MHCI and MHCII surface expression by MΦ exposed to TbEVs. After 24 h of incubation, the frequency of MHCI^+^ (**A**), MHCII^+^ (**B**), and MHCI^+^MHCII^+^ cells (**C**) was evaluated in MΦ exposed to *T. b. brucei* (*Tbb*) parasites, in MΦ stimulated by TbEVs and PMA, and in resting-MΦ (negative control, NC). Results of at least three independent experiments (*n* = 32) and at two replicates per sample are represented by whisker box-plot, median, minimum, and maximum values. The non-parametric Wilcoxon matched-pairs signed-rank test was used for statistical comparisons. * (*p* < 0.05), ** (*p* < 0.01), *** (*p* < 0.001), and **** (*p* < 0.0001) indicate statistically significant values.

**Figure 6 biomedicines-09-01056-f006:**
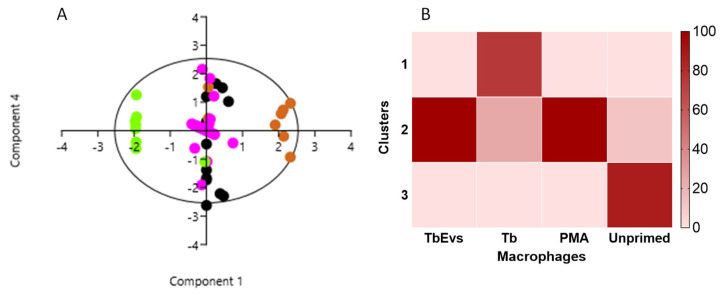
Influence of TbbEVs on mice macrophages. The relationship between the effect of TbbEVs (black dots), PMA (positive control, pink dots), and *T. b. brucei* parasites (brown dots) in MØ, including urea and NO production and density of MHCI and MHCII, is represented by a variable correlation plot (**A**). Resting MØ (negative control) are indicated by green dots. Cluster distribution of the effect TbEVs, PMA, and parasites (Tb) in MØ compared with resting cells (NC) are represented by heat map (**B**).

**Figure 7 biomedicines-09-01056-f007:**
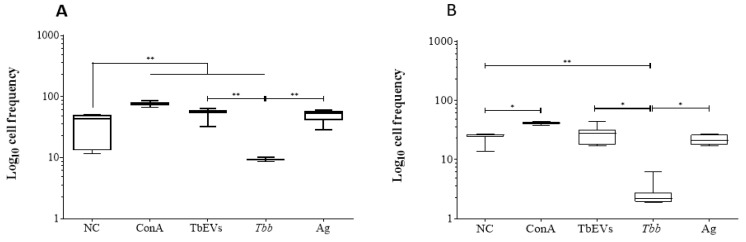
Frequency of CD3^+^ cells after exposure of mononuclear cells to TbEVs. CD4^+^ (**A**) and CD8^+^ (**B**) cell fractions exposed to *T. b. brucei* parasites (*Tbb*), stimulated by parasite antigen (Ag), TbEVs, and ConA, and unprimed cells (negative control, NC) were labelled with CD3-monoclonal antibody and evaluated by flow cytometry. Results of at least three independent experiments (*n* = 20) performed in duplicate are represented by whiskered box-plot, median, minimum, and maximum values. The non-parametric Wilcoxon matched-pair signed-rank test was used for statistical comparisons. * (*p* < 0.05) and ** (*p* < 0.01) indicate significant differences.

**Figure 8 biomedicines-09-01056-f008:**
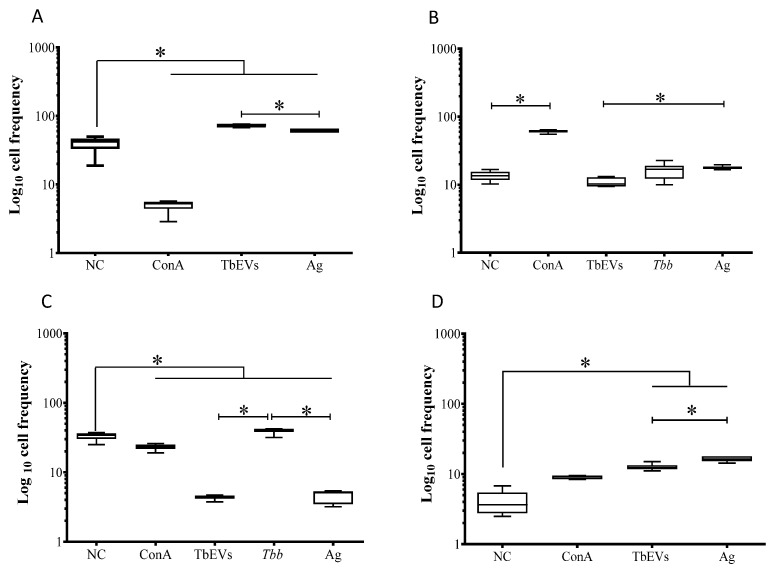
Differentiation of CD25^+^ and FoxP3^+^ CD4^+^ T cell subsets induced by *T. b. brucei* EVs. CD4^+^ cell fraction exposed to *T. b. brucei* parasites (*Tbb*), stimulated by Ag, TbbEVs, and ConA, and unprimed cells (negative control, NC) were labelled with CD3, CD25 and FoxP3 monoclonal antibodies and evaluated by flow cytometry. CD3^+^ cells were gated and the frequency of CD25^+^FoxP3^+^ (**A**), CD25^−^FoxP3^−^ (**B**) CD25^+^FoxP3^−^ (**C**) and CD25^−^FoxP3^+^ (**D**) cell subsets were estimated. In consequence of the low levels of CD3^+^ cells, the frequency and density of FoxP3 were not considered in *T. b. brucei* exposed-PBMC. Results of three independent experiments (*n* = 18) performed in duplicate are represented by whiskered box-plot, median, minimum, and maximum values. The non-parametric Wilcoxon matched-pair signed-rank test was used for statistical comparisons. * (*p* < 0.05) indicated significant differences.

**Figure 9 biomedicines-09-01056-f009:**
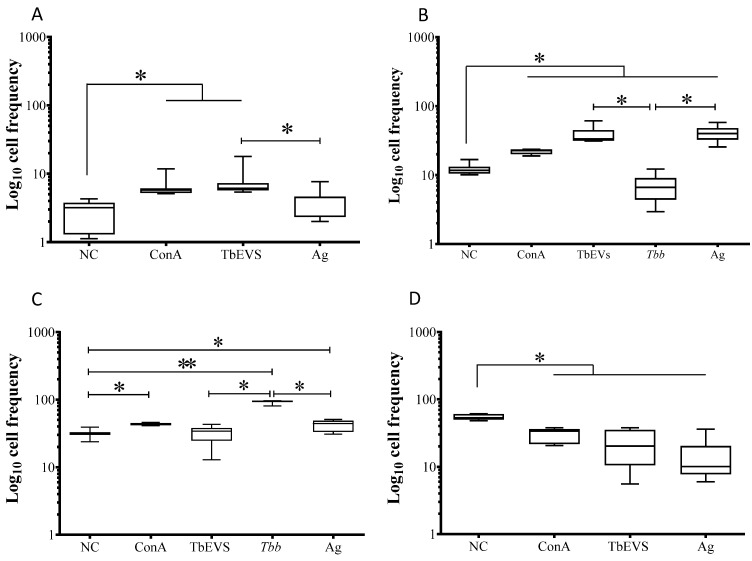
CD25^+^ and FoxP3^+^ CD8^+^ T cell subsets induced by *T. b. brucei* EVs. Mouse lymphocytes exposed to *T. b. brucei* parasites (*Tbb*), stimulated by Ag, TbEVs, and ConA, and unprimed cells (negative control, NC) were labelled with CD3, CD25 and FoxP3 monoclonal antibodies and evaluated by flow cytometry. CD3^+^ cells were gated and the frequency of CD25^+^FoxP3^+^ (**A**), CD25^−^FoxP3^−^ (**B**), CD25^+^FoxP3^−^ (**C**), and CD25^−^FoxP3^+^ (**D**) cell subsets were estimated. In consequence of the low levels of CD3^+^ cells, the frequency and density of FoxP3 were not considered in *T. b. brucei* exposed-PBMC. Results of (*n* = 6) at least three independent experiments performed in duplicate are represented by whiskers box-plot, median, minimum and maximum values. The non-parametric Wilcoxon matched-pair signed-rank test was used for statistical comparisons. * (*p* < 0.05) and ** (*p* < 0.01) indicate significant differences.

**Figure 10 biomedicines-09-01056-f010:**
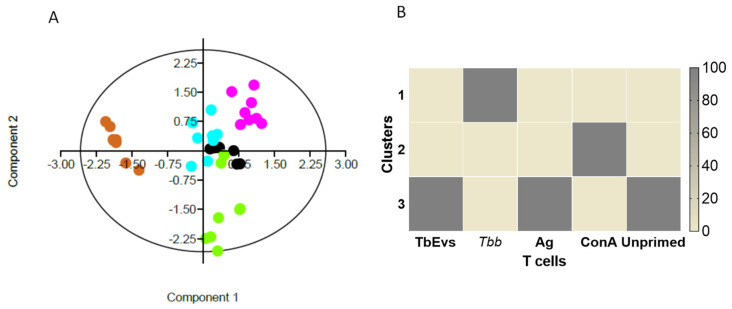
The effect TbEVs on mice T cells. Relationship between the effect of TbEVs (black dots), parasite Ag (blue dots), ConA (pink dots) and viable *T. b. brucei* trypomastigotes (brown dots) in CD3^+^ cells, including frequency of CD25^+^, FoxP^+^, and CD25^+^FoxP3^+^ subsets and the density of CD3, CD25, and FoxP3 molecules are represented by a variable correlation plot (**A**). In consequence of the low levels of CD3^+^ cells, the frequency and density of FoxP3 were not considered in *T. b. brucei* exposed-PBMC. Unprimed cells (negative control) are indicated by green dots. Cluster distribution of the effect TbEVs, parasite Ag, ConA, and trypomastigotes (*Tbb*) on T cells compared with unprimed cells (NC) are represented by the heat map (**B**).

## Data Availability

The data presented in this study are available on request from the corresponding author (G.S.-G.). The data are not publicly available due to confidentiality.

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
