# Peer review of "African Trypanosomiasis: Extracellular Vesicles Shed by *Trypanosoma brucei brucei* Manipulate Host Mononuclear Cells"

_biomedicines, 2021, doi:10.3390/biomedicines9081056_

Round 1
Reviewer 1 Report
In this study, the effect of extracellular vesicles released from T. b. brucei (TbEVs) on macrophages and T cells was examined since the role of TbEVs in host immune response remains unclear.
- Overall, the objective is unclear, and many results were hard to understand.
- Introduction is too long with many unnecessary descriptions. It should be more focused.
- There are many unnecessary figures. For examples, Figs. 1, 6, 7, 9, 11, 12, 14, and 15.
- I could not understand the results of Figs. 5 and 6. What is MHCI- and MHCII- MΦ? All nucleated cells are MHCI+. In addition, MΦ are MHCII+ cells, although the expression level is different among subpopulations.
- What does “down expression of CD3” (line 422) mean? Decrease in the number of CD3+ cells? Or down regulation of CD3?
- CD4+CD25+FoxP3+ cells are defined as regulatory cells, but not suppressive cells.
- Line 426: What does “differentiation” mean? Do you mean CD4-CD8- (double negative) T cells differentiate into single positive T cell by TbEVs?
- “Stimulation of PBMC by ConA led to a significant expansion (p = 0.0313) of FoxP3+ in CD4+ T cell and FoxP3+CD25+CD4+ T cell subsets (lines 437-438).” Generally, ConA stimulation expands effector T cells, but not CD4+CD25+FoxP3+ regulatory T cell population.
- T cells (CD3+) were markedly decreased when exposed tob. brucei (Fig. 8). However, T cells subsets such as CD4+CD25+FoxP3+ were compared in later studies (Figs. 10-14). I wonder T cell subpopulations can be compared under such biased conditions.
- Macrophages were abbreviated sometimes as MΦ and other times as MF.
- Species names should be written in italic.
Author Response
The authors of the current manuscript acknowledge the time and effort of the referee in reviewing our study.
I - Reply to reviewer 1
Questions and comments
- Overall, the objective is unclear, and many results were hard to understand.
- Introduction is too long with many unnecessary descriptions. It should be more focused.
A: Thank you for both comments. The introduction was shortened and more focused and the objective of the study was clarified. Moreover, data description was improved to facilitate the understanding of the effect of TbEVs in macrophages and T cells.
- There are many unnecessary figures. For examples, Figs. 1, 6, 7, 9, 11, 12, 14, and 15.
A: The authors would like to maintain the following figures:
Fig. 1 – This figure shows the trypomastigote morphology of mice parasites in culture, which indicates that EVs purified from culture supernatant and used in the current study were shed by T. b. brucei trypomastigote forms. This is important because with time in vitro T. brucei wild type parasites become unable to replicate, lose their motility and change morphology.
Figures 7 and 15 (that are now figure 6 and 10) – Using two different analytic methodologies, these figures allow us to visualize that the effect of EVs on mouse macrophage and T cells does not match the effect of T. b. brucei trypomastigotes on that same cells. The authors followed the recommendation of reviewer 2 and a brief explanation for the application of PCA analysis has been introduced in M&M section (2.10. Statistical analysis), improving the understanding of this graphic representation of data.
Regarding figures 6, 9, 11, 12 and 14, the authors agree with the reviewer that these figures are not essential to the understanding of this study and were deleted.
.
- I could not understand the results of Figs. 5 and 6. What is MHCI- and MHCII- MΦ? All nucleated cells are MHCI+. In addition, MΦ are MHCII+ cells, although the expression level is different among subpopulations.
A: The authors agree with the reviewer that most mouse somatic cells express constitutively MHCI due to a conserved cis-element in the upstream region of MHC class I genes (MHC-CRE) that that can mediate the up-regulation of MHCI molecules. Moreover, CIITA regulates the transcription of MHCII molecules and improves the constitutive MHC class I gene expression. In this study, a mouse lymphoid neoplasm P388D1 cell line with monocyte/macrophage characteristics was used to evaluate the expression of MHC when stimulated by TbEVs. This cell line has a low expression of MHC and at least 25% of resting cells are MHCI/MHCII negative. Thus, TbEVs enhance MHC expression and T. b. brucei parasites do not. It is well demonstrated that parasites can download or even abolish the expression of MHC molecules, avoiding antigen presentation as a survival strategy.
- What does “down expression of CD3” (line 422) mean? Decrease in the number of CD3+ cells? Or down-regulation of CD3?
A: The words up- and down-regulation are usually applied to the transcriptome (Exp: gene expression) whereas over and down expression are used for the phenotype (Exp. CD3 expression). So, we think that should be “down-expression” of surface molecules.
- CD4+CD25+FoxP3+ cells are defined as regulatory cells, but not suppressive cells.
A: Immunosuppressive cells were replaced by regulatory cells. Thanks.
- Line 426: What does “differentiation” mean? Do you mean CD4-CD8- (double negative) T cells differentiate into single positive T cell by TbEVs?
A: CD4+ and CD8+ T cell populations differentiate within the thymus and are believed to be stable, being able to mature and proliferate after the antigenic presentation. Although some studies support the reexpression of these co-receptors, in this case, we aim to evaluate the effect of TbEVs on CD4+ and CD8+ T cell populations. Thus, this sentence was changed.
- “Stimulation of PBMC by ConA led to a significant expansion (p = 0.0313) of FoxP3+ in CD4+ T cell and FoxP3+CD25+CD4+ T cell subsets (lines 437-438).” Generally, ConA stimulation expands effector T cells, but not CD4+CD25+FoxP3+ regulatory T cell population.
A: Expansion of foxP3+ T cells by ConA stimulation has been refereed in the literature and seem to be ConA dose dependent and eventually are associated with the possible expansion of natural Tregs. However in this case, the legend of Fig. 10 was inaccurate which led to an incorrect description. The legend and the respective data description have been changed.
- T cells (CD3+) were markedly decreased when exposed to brucei(Fig. 8). However, T cells subsets such as CD4+CD25+FoxP3+ were compared in later studies (Figs. 10-14). I wonder T cell subpopulations can be compared under such biased conditions.
A: We agree with the reviewer comment and the results regarding FoxP3 of cells exposed to parasites were eliminated from the manuscript.
- Macrophages were abbreviated sometimes as MΦ and other times as MF.
A: Macrophage abbreviation was uniformized to MØ.
- Species names should be written in italic.
A: All specific names are now in italic.
Reviewer 2 Report
Please see attached file for comments. Thank you.
Author Response
The authors of the current manuscript acknowledge the time and effort of the referee in reviewing our manuscript.
Major Comments
Abstract
- In the abstract the authors state that “African trpyanosomosis or sleeping sickness is a zoonotic disease caused by Trypanosoma brucei brucei…” This is not exactly correct given that Trypanosoma brucei brucei is the causative agent of animal infection but rarely transmitted to humans, the definition of zoonotic. The other subspecies of Trypanosomes brucei rhodesiense and T. brucei gambiense are the causative agents of disease in humans with animal reservoirs. Therefore the authors should be careful about how they word this section. Also it should be African trypanosomiasis not trypanosomosis.
A: Thank you very much for your pertinent comment. The abstract was corrected for Trypanosome brucei
- Introduction
- In the introduction the authors mention there are limited treatments many of which have safety concerns and there are no vaccines available but this section would also benefit from some epidemiological information regarding incidence and prevalence to show why it is necessary to better understand the immunology of the disease.
- Following reviewer 1 comments the introduction was deeply modified and focused on the manuscript main subject. Thus, references to treatments and vaccines were deleted.
- Materials and methods
- In section 2.1. Experimental Design the authors should consider writing the first paragraph in a different format. Instead of saying the study had three main steps for the
design the authors should consider describing the steps as specific aims of their study.
A: The first paragraph of Experimental Design (subhead 2.1.) was changed following the reviewer comments.
- In the statistical methods section the authors should further explains why the non-parametric Wilcoxon-matched-pair-signed rank test was appropriate. Were any statistical tests used to assess normality? Furthermore the authors should also discuss why the PCA was an appropriate analysis and any assumptions that were made for that analysis. Lastly, the authors used a software called Past4 and should include what the software’s full name is along with location of the company same for Graph Pad Prism version 8 (San Diego, CA, USA).
- Data normality was analysed with the Kolmogorov–Smirnov Since our data did not present a normal distribution was analysed using the non-parametric Wilcoxon-matched-pair-signed rank test for related samples. Principal Component Analysis (PCA) was applied to organize data in principal components and can be visualized graphically with minimal loss of information and make visible the differences between groups (cells stimulated by TbEVs, Ag, ConA, cells exposed to parasites, and resting cells) which enhance data understanding. For this, data normality is not a requirement.
- The authors should include the sample size for all experiments. It is not appropriate to just say three independent experiments and at least two replications per sample. Please state the exact sample size (N) for each experiment and analysis.
- Since sample size presents some variation, the exact number of samples evaluated was included as appropriate in the figure legends.
Minor Edits:
Overall reduction
- The authors need to italicize Trypanosoma brucei brucei throughout the entire manuscript. At this time it is not italicized at all.
- All the specific names are now in italic
- The authors should review the entire paper for spacing errors. I noticed at lines 28, 33, 36, 333, 336, etc. there are too many or too few spaces. The spacing between paragraphs also results in issues with titles being listed at the bottom of a page this should be addressed.
- The spacing errors and space between lines and paragraphs were revised.
Title and Authors List
- The authors should fix the superscript in the author list for the 1st, 2nd, and 6th
- The superscript identifier numbers were corrected.
- At line 28 it should read “…in host immune response remains poorly understood.”
- At line 36 it should read “…factor FoxP3 lead to the differentiation of…”
- A. The authors deeply acknowledge the reviewer suggestions that were introduced in the abstract.
- Introduction
- At line 50 it should read “Since safe and effective anti-trypanosomal drugs and vaccine are lacking many studies have been performed to understand the mechanisms associated with the host’s direct immune response against T. brucei infection. Although different approaches have been applied, and numerous findings reported, some questions still remain unanswered, and some mechanisms are not entirely understood or are still controversial.”
- At lines 126-127 and at line 131 the symptoms for INF-ƴ and NF-κB are not appearing in the text.
- After the introduction reformulation, the text mentioned above was deleted.
Figures
- In figures 7 and 12 the heat map should also be in color.
A: The heat maps are now in colours
- For figures 6, 9, 11, and 14 the authors should adjust the y-axis since the plots are difficult to view. The authors may consider starting the y-axis at 10 instead of 1.
A: Thank you very much for the suggestion, but we followed the reviewer 1 recommendation and these figures were deleted.
Round 2
Reviewer 1 Report
The manuscript was revised properly.